# Pilot Testing an Ecotherapy Program for Adolescence: Initial Findings and Methodological Reflections

**DOI:** 10.3390/ijerph22050720

**Published:** 2025-05-01

**Authors:** Sophie Westwood, Grace Edmunds-Jones, Thomas Maguire, Sue Hawley, Hannah Avent, Jerry Griffiths, Rishi Bates, Jane Marley, Gary Wallace, Ruth Harrell, Sheena Asthana, Felix Gradinger

**Affiliations:** 1Community and Primary Care Research Group, National Institute of Health and Care Research (NIHR) Health Determinants Research Collaboration (HDRC) Plymouth, Faculty of Health, University of Plymouth, Plymouth PL4 8AA, UK; 2Public Health, Plymouth City Council, Plymouth PL1 3BJ, UK; 3Street Services, Environmental Planning, Natural Infrastructure Team, Plymouth City Council, Plymouth PL1 3BJ, UK; 4Child and Adolescent Mental Health Services (CAMHS), Livewell Southwest Community Interest Company, Plymouth PL4 7PY, UK; 5Improving Lives Plymouth, Changing Futures Plymouth, Community Connections, Plymouth City Council, Plymouth PL1 3BJ, UK; 6Youth Services, Community Connections, Plymouth City Council, Plymouth PL1 3BJ, UK; 7Peninsula Medical School, Faculty of Health, University of Plymouth, Plymouth PL4 8AA, UK

**Keywords:** child and adolescent mental health, ecotherapy, green/blue (social) prescribing, nature—based approaches (NBAs), nature connectedness, young people

## Abstract

Children and young people’s mental health and well-being has seen a dramatic decline. In the UK, this has been exacerbated by service retrenchment associated with austerity, with evidence of increasing health inequalities. Service innovation that is grounded in practice, has ongoing learning, and is co-designed with children and young people is required now. This can provide creative solutions within the local context and contribute to the fledgling evidence base that explores complex mechanisms of impact. This methodological reflection describes a co-design process of a bespoke, group-based ecotherapy programme: from early piloting using appreciative enquiry before COVID-19 by the mental health, public health, and Street Services team in the port city of Plymouth, to further developing an evaluation framework through an innovative, matched-funded academia–practice partnership. The findings showcase the benefits of a systems-based approach to public, multi-agency and academic collaboration, facilitated by peer and practitioner researchers and embedded researchers-in-residence. They highlight the need to consider nuances of specific (connecting with self, others, animals, nature) and non-specific active ingredients of the emerging and constantly adapting service (therapeutic relationship with practitioners/carers; nature as therapist, and group dynamics), as well as the value of pragmatic and participatory evaluation methods (distance-travelled, goal-based measures; and ethnographic, qualitative observation), to provide rapid, continuous, and real-time learning and improvement.

## 1. Introduction

Just as glaciers melt away under the acceleration of the climate crisis, the mental health of young people is being worn down by the wider mental health crisis and inadequate service provision. What used to be a dip followed by recovery in mid-life (the traditional ‘hump shape’), is now turning into a constant erosion over time to a “monotonic decrease in illbeing by age”, as evidenced in the April 2024 working paper by the US National Bureau of Economic Research [1]. The UK finds itself at the very bottom of 71 countries surveyed and ranked in terms of mental well-being in the most recent Mental State of the World report, published in March 2024 [2]. In April 2024, 30% of children in the UK were living in poverty [3], estimated to give a rise in costs to the system of £40 billion a year by 2027 [4]. One in three mental health problems in adulthood is directly connected to an adverse childhood experience (ACE) [5], highlighting the rise in multiple and complex needs requiring multiple agencies and services to work together. In 2023, about one in five children and young people in the UK aged 8 to 25 years had a probable mental disorder, a huge increase from one in nine in 2017 [6]. COVID-19 has exacerbated the record-breaking rising demand and waiting lists for Child and Adolescent Mental Health Services (CAMHS) in the UK [7], with a 53% increase over the past four years in the number of children in mental health crisis who need emergency support [8]. A programme of austerity measures since the 2008 financial crisis has decimated public services in the UK, with Youth Services being cut by 73% since 2010 [9].

A 2021 UNICEF discussion paper highlights the necessity of urban green space for children’s development [10]. Big data explorations in the UK showcase the protective aspect of access to green spaces during COVID-19 [11], while this was unequally patterned in terms of socio-economic status and protected characteristics [12]. More generally, there is strong longitudinal evidence to suggest the benefits of connecting with nature across a range of associated outcomes, including brain development [13], cognitive abilities [14], social-emotional learning skills [15], behaviour [16], and physical health (e.g., immune system [17] and bone density [18]). This particular field of research is open to furthering current knowledge gaps regarding qualitative process indicators, theory around mechanisms of change, and consideration of wider outcome measures.

Human Learning Systems (HLS) “https://www.humanlearning.systems/ (accessed on 20 April 2025)” and system stewardship using an embedded researcher-in-residence [19] approach have been proposed as ways of delivering creative solutions within local contexts, facilitating more immediate improvement (do–know gap) rather than waiting for an average of 17 years for traditional evidence creation to feed into policy decisions (know–do gap) [20]. HLS emphasises the need to progress towards the more systematic collection of relevant and timely formal and informal knowledge to inform and drive improvement in service delivery; to change the culture of diverse and disparate organisations and stakeholders towards more reflective and iterative learning practice; and to develop the skills to achieve this and translate a better understanding of what works into sustainable improved practice in order to harness the preventative benefits over a lifetime.

Given the crisis in mental health among children and young people (CYP), the notion that the public and Voluntary and Community, Faith and Social Enterprise (VCSFE) sector services need to be liberated from a siloed approach [21] to service provision to deliver bespoke, inclusive support catering to the varying needs and strengths of CYP is compelling. The recent UK policy drive to invest in social prescribing programmes, mainly for adults [22], has led to an interest in providing access to ‘nature-on-prescription’, often linked to VCSFE sector offers, in lieu of public service activity. However, the aspects of why and how this might work for CYP in terms of more general [23] or nature-specific [24] mechanisms, how to make this equitable and inclusive [25], or how best to implement and adapt interventions in context [26,27] remain under-researched.

The challenges here extend to defining and agreeing on concepts like nature connectedness (an individual’s subjective sense of their relationship with the natural world) and how to measure it [28]; how disconnections from nature may intersect with the wider social and environmental determinants of health [29]; and what this means for capacity to benefit. Furthermore, contextually rich, methodological, and practice-based evidence of what is actually meaningful to young people, parents, and carers themselves is scarce [30].

The purpose of this paper was to (a) provide preliminary findings from an early internal pilot and service evaluation; (b) report the methodological reflections made by the team that informed the co-design and implementation of the subsequently improved service offer; and (c) discuss the implications for embedded, co-produced learning approaches in this field and more widely. This paper reports on a small-scale pilot and internal ecotherapy service evaluation, providing an example of an innovative co-design and evaluation process of a group-based ecotherapy service for 11- to 18-year-old CYP (up to 25 years if living with Special Educational Needs) with mild to moderate mental health issues (anxiety and depression). It seeks to showcase how embedded, iterative learning across agencies/disciplines/professions and in partnership with practitioner researchers, peer researchers, and researchers-in-residence can provide rapid, continuous, and real-time service development, innovation, and improvement in the relevant and rigorous middle ground between clinical audit, evaluation, and research.

## 2. Materials and Methods

### 2.1. Internal Pilot and Service Evaluation Before/During COVID-19

This work is set in the Southwest Peninsula of the UK, a largely rural and mixed urban area of outstanding natural beauty with a mixed economy of tourism [31], and it is a favoured retirement destination with a rapidly ageing population, huge inequalities, and significant pockets of coastal deprivation [32], with huge implications for raising families who often do not have the opportunity to benefit from accessing nature [33]. The public health team at Plymouth City Council (PCC), UK, were involved in early pilot work at the beginning of COVID in partnership with Street Services (a sub-directorate of which is Environmental Planning, and within this the Natural Infrastructure team related to managing public parks, etc.), exploring, through qualitative appreciative enquiry [34], the impact of nature-based experiences on CYP and their parents at PCC’s Poole Farm (green prescribing; https://www.plymouth.gov.uk/poole-farm, accessed on 20 April 2025)). This provided an initial evidence base to help secure Youth Investment Funding (YIF) to support a future two-year ecotherapy pilot comprising an implementation arm through a collaboration between the mental health service provider Livewell, the Environmental Planning team (Street Services, Place Directorate) and Youth Services (Community Connections, Adults, Health and Communities Directorate), as well as an evaluation co-design framework developed in collaboration with the Plymouth NIHR Health Determinants Research Collaboration (https://www.plymouth.gov.uk/plymouth-health-determinants-research-collaboration-phdrc, accessed on 20 April 2025). The fuller evaluation will be reported in 2025/26.

Convenience sample: The initial internal service evaluation comprised three cohorts between December 2021 and June 2022, all conducted at a single site (green prescribing at Poole Farm), a time impacted by the COVID pandemic. All cohorts were recruited through Child and Adolescent Mental Health Services (CAMHS) and included children with low-level mental health issues or those who had benefited from previous therapy in the service. After the eight weeks of the service, it was difficult to follow up more robustly and face-to-face with the first cohort as planned because there were concerns around the COVID pandemic. Instead, Appreciative Enquiries were conducted on MSTeams as a more open, less intrusive, and more accessible approach. The first two cohorts of six and eight children were a mix of girls and boys between the ages of 8 and 14 years. The third cohort was a larger group of ten boys aged 8 to 13 years, with a higher proportion of neurodiversity.

Mixed methods and mixed data collection and analysis: The following prompts were used for qualitative data collection, which was recorded manually using field notes (where face-to-face) or recorded on *MSTeams* (online during lockdowns). The transcripts were non-systematically analysed, using basic content and thematic analysis, conducted by non-academic staff members (JM, HA, GEJ, GW—producing two separate ‘Outside-Within’ evaluation reports). The condensed write-up presented here was facilitated by the first author:

Appreciative enquiry prompts directed at the CYP: Tell us about your family; Tell us about your experiences of Poole Farm; What’s your favourite green space?; How does being in green spaces make you feel?; Tell me about something you learnt about at Poole Farm.

Appreciative enquiry prompts directed at the parents/carers: Tell me about your family; Tell me about your favourite green space; How’s it been to share your experience with others?; What do you think your child has gained from the sessions at Poole Farm?

Appreciative enquiry prompts directed at both CYP/parents/carers: Tell me about your typical day; Tell me about your best day; Tell me about your nightmare day; Tell me about your favourite part of the outside within the programme.

In addition to the qualitative insights gathered through appreciative enquiry, the team also collected as much quantitative data from the third cohort as possible, given the pragmatic constraints described above. The team used the Revised Child Anxiety and Depression Scale (RCADS) [35], which is a 47-item, youth self-report questionnaire with subscales including separation anxiety disorder, social phobia, generalised anxiety disorder, panic disorder, obsessive–compulsive disorder, and low mood (major depressive disorder). It also yields a Total Anxiety scale (sum of the 5 anxiety subscales) and a Total Internalising scale (sum of all 6 subscales). Findings here report on basic before and after percentage changes for the Total Anxiety and Depression scores.

### 2.2. Ecotherapy Co-Design Events

In the context of public service cuts, the natural infrastructure team is successfully entrepreneurial and largely self-funded through bidding for external sources. This includes using the initial findings from the above pilot to help secure Youth Investment Fund revenue (largely capital to develop the sites, but some for services like this). Once the therapist was recruited in Dec 2023, the co-design of the service specifications for a more formalised 6-session protocol started, and the service has been running fuller developed groups at Poole Farm from April 2024 (3 sessions a week with 4–6 young people either 11–13, or 14–18 or 25 if there are additional special education needs). This process drew in partners from various relevant parts of the system. It was facilitated by the Livewell participation and engagement lead, the practitioner researchers seconded by the HDRC to work part-time with the researchers-in-residence. Furthermore, initial feedback from one of the first groups run was collected by a young people peer researcher from Changing Futures Plymouth (https://theplymouthalliance.co.uk/changing-futures, accessed on 20 April 2025). Two participation events were held, with young people recruited through an advertising campaign within Youth Services. Four young people attended in February and nine in March 2024. There was a mix of males and females, non-binary and ‘prefer not to say’, aged 11–17 years.

The Ecotherapy Lead (a mental health nurse familiar with facilitating CYP groups) led the sessions with the help of the Natural Infrastructure (Street Services) Assistant, a ‘practitioner researcher’ seconded to HDRC. An HDRC partner (researchers-in-residence, two University researchers seconded to the public health team) attended each session, and the Livewell participation lead led the February session. The 3 carers of the February group also attended; two youth workers attended with the March group.

At the sessions, the Ecotherapy Lead explained the idea behind the project. The participants were shown around Poole Farm, during which they could feed and touch the farm animals (ducks and pigs), and they took part in an ecotherapy taster activity (fox-walking, including being blindfolded if comfortable to do so). Fox-walking offers a choice of activities that include playfully pretending to enact and use a fox’s senses, either self-led or by another group member. The purported mechanisms are around mindfulness, attuning senses other than sight, and trusting self and others. Participants shared their reflections on the project around the fire (with snacks/marshmallows). Various insights came to light, which led to the iterative reshaping of some project activities (see below). The decision was also made to embed formative evaluation through less intrusive in-session observations (use of less formal self-reported outcome measures) throughout to support a continuous circle of learning.

## 3. Results

The learning has been iterative, and the following combines the findings of the initial appreciative enquiry evaluation and subsequent co-design. We begin by considering key themes that emerged through interviews and observations from the Section 2.1.

### 3.1. Relationships and Connection with Other People

The group aspect of the service was found to be important for combating loneliness, helping participants to normalise or accept their experiences, and becoming open to help from others. The participants’ favourite memories included meeting and making friends through the group, and participants also reported making friends with peers elsewhere more easily. Parents felt their children had become more open with them and their peers outside of sessions. One parent reflected on how walking in green spaces helped their child express their feelings. Facilitators observed the children actively sharing with one another, working together as a team, and considering each other’s needs, demonstrating their connection. Learning to make and feel confident in connections with others is a new skill for some of these children, and they found that they could cope better with daily challenges.

### 3.2. Enjoyment and Fun

There was no attrition, and both parents and children talked about both the activities and the time spent with peers as fun and enjoyable. As a result, the children engaged well with the sessions and were not self-conscious about their learning: it was educational by stealth. The efficacy of learning about mental health and resilience in an uplifting and joyful way comes out throughout the data and is a key quality of this programme. So, while children were being challenged to step outside their comfort zones by disrupting their routines and taking responsible risks with the support of others, they were finding that they could cope better than usual and enjoyed succeeding; therefore, they grew in confidence, felt a sense of achievement, developed a ‘can do mindset’, and associated enjoyment with being nervous or even scared in a supported and safe environment.

### 3.3. Improved Confidence and Independence

The children learnt new practical, social, and emotional skills, all of which contributed to an improvement in their confidence and independence. Within the groups, the children found their own roles; for example, older children caring for younger children. Some children benefited from time away from their siblings, providing them with the space needed for their own self-development. Both the children and their parents noticed how they grew in confidence to speak with other people and be in groups. This learning was transferred to school, including qualities of resilience and persistence in more challenging classes.

### 3.4. Being in the Outdoors

Being outside and surrounded by nature helped the children to relax and engage with the sessions, it gave them a greater sense of freedom, and it was discernibly different from school. The children also really enjoyed interacting with the animals at Poole Farm, caring for them and finding themselves calmed by them. It was recognised that social interaction is easier outside. Being outside makes the children feel good, talk more openly, and manage difficult thoughts more effectively.

When reflecting on the outside spaces they usually visit, most of the participants talked about ordinarily being outside when taking part in a sporting activity; only one child talked about spending time or walking in nature. The Outside Within programme had helped some of the children to think about where they spend their time.

### 3.5. Wider Impacts

Parents attributed their child’s improved engagement at school and dealing with unforeseen events in daily life to the ‘Outside Within’ programme. Because of the social and emotional skills they gained, the children learnt to feel more confident in taking reasonable and responsible risks and had developed a willingness to give things a go, to cope with unexpected situations, and, in general, to be more confident in stepping outside their comfort zones.

### 3.6. Revised Child Anxiety and Depression Scale (RCADS) Scores

In addition to collecting information from interviews and observations, the Revised Child Anxiety and Depression Scale (RCADS) [35] scores were collected before and after the third cohort that was involved in the internal service evaluation. This comprised eight boys and/or their parents. While there were male participants only, and it was a very small sample, the results remain promising if not slightly inconclusive (see Table 1). Five children start beneath the clinical threshold of a T Score of 65 (no referral to treatment indicated, unless clinical judgment suggests otherwise). One sat on the borderline (65–69), and two were in clinical range (T Scores of 70 or above). Modest improvements were observed for five of the children. One child did not improve, and another dropped meaningfully in scores.

### 3.7. Co-Design of the Ecotherapy Service and Evaluation Framework in 2024

Both co-design events described in Section 2.2 were received positively, with the young people enjoying the activities, interaction with the animals, the natural setting, and freedom of movement. There was some trepidation about having to fill in forms when offering the visitors a range of candidate questionnaires, which was subsequently discussed among the team. Also discussed were the positive impact of the sensory experiences, opportunities for bonding, taking responsibility, testing or reducing anxiety, and the case for building these components into the planned sessions and subsequent evaluation through more observational, ethnographic methods [36].

At the March 2024 co-design event, some additional considerations came to light, such as having waterproofs and wellies in the appropriate sizes to loan to participants and awareness that the concerns of the young people may differ from those perceived by their carers, who may potentially be the ones making the referral. It was felt that a smaller group was more productive for various reasons. In terms of the balancing of heterogeneity and homogeneity of in-group dynamics, the larger group showed divisions between louder/quieter young people and different sub-groups that knew each other. Running sessions for groups from schools/settings was identified as a good idea, both for recruitment and organisational purposes and for group dynamics during the sessions. The challenges of meeting the needs of neurodiversity, as well as the therapeutic needs of emotional and mental health needs, were highlighted. Behaviour management seemed easier and more informal (less school-like) in smaller groups, which also gives more opportunity for the therapeutic relationship between each young person and the leaders to develop. Transport was identified as a challenge for young people, especially given levels of anxiety, and the associated costs of using taxis (some covered by Livewell Foundation).

Continued iteration occurred as the service rolled out, session plans being adjusted as it became clearer what worked and what did not. For example, during the co-design events, the Ecotherapy Lead had noticed how much the young people had enjoyed toasting marshmallows around the fire, something that had become a ritual at the end of sessions—also conveniently used to gather feedback and debrief informally. In time, it became apparent that this activity was starting to dominate the sessions, with some young people unable to fully engage because they were anticipating the ritual and then hyper-focusing on the activity. Others felt uncomfortable with the social aspect, not yet feeling confident to talk within the group—suggesting that considerable grounding and gelling as a group should form part of the initial session. The Ecotherapy Lead responded by limiting the activity and changing the format of the sessions, without losing the elements that had been helpful: the challenge of fire making, fun, and socialising as a group. All of these will add to further lines of enquiry in future evaluations, including topics like previous and new experiences of ritual, routine, and boundaries. Similarly, during the co-design sessions, it emerged that it was important to be clear about expectations for behaving appropriately around the animals, so that they were not stressed by too many enthusiastic young people. ‘Fox-walking’, including being blindfolded, proved to be very effective, providing the right balance between challenge, fun, social interaction, mindfulness, and sensory exploration of nature. It has become a signature activity used with work experience students and other visiting groups to showcase what the project involves.

Furthermore, ‘on-boarding’ of the referral process continually evolved after the co-design events in 2024, including going to significant lengths in having one-to-one grounding and settling support, taster visits with parents/carers, and plans to offer parents an opportunity for peer support while CYP visit the farm enclosure. Feedback from some parents and CYP included the rejection of the word ‘therapy’ in ecotherapy, emphasising the loaded and possibly stigmatising nature and that this starts from a narrow, clinical, and deficit-based mental health model, when the experiences offer so much more than symptom reduction. This has not been resolved, as we continue to puzzle about a way to capture all aspects and all audiences. After the co-design events and since 24 June/July, parents and the ecotherapy groups can also use a room in the newly refurbished ‘Hayloft’ as their meeting and ‘launch’ point before exploring the farm/woods, which has been carefully refurbished to give a neutral ‘feel’ and ensure the surroundings are non-institutional/non-clinical.

In terms of sustainability and the portability of enhancing green awareness more widely, the team benefited from a strong collaboration with the *National Trust* (https://www.nationaltrust.org.uk/who-we-are/our-strategy; accessed on 20 April 2025; one of the largest landowners and heritage and nature conservation charities in the UK), who have developed marketing materials (‘spirit of place’), have put in place a partnership role with the Council, and provide holiday activities, etc. The team further developed ways for people to continue having a relationship with the farm and other green and blue spaces through numerous other activities (e.g., tree growing/planting for the City; ‘Wild and Well’ one-off events; National Marine Park (https://plymouthsoundnationalmarinepark.com/, accessed on 20 April 2025); and Green Minds neighborhood-based work (https://greenmindsplymouth.com/; accessed on 20 April 2025, volunteering, etc.). This includes the option of young people ‘graduating’ from ecotherapy to sign up for very popular, open evening drop-ins of up to 25 people, facilitated by the Youth Services team.

A more technical change was the decision to replace the RCADS with the MyCAW [37] tool to collect baseline and end-of-service data. We noticed that young people were not keen on completing paperwork at the co-design sessions. For some, it reminded them of school content, triggering learning difficulties and some anxiety around providing personal information in a group scenario. The team discussed respecting CYP’s views about long and possibly intrusive routine outcome measures and ways in which to manage the difficulties in collecting written information on the young people’s needs and progress. We agreed with CYP that the MyCAW tool was well designed and simpler for young people to complete. For example, this tool is goal-oriented and person-led, prompting young people to think about and articulate their two main concerns and rating the extent to which they are important now, as well as rating their overall well-being. As an exercise, this proved useful for providing the young people with a purpose for coming to the sessions and suggests some of the issues that the group facilitators are there to support them with.

Embracing the need for ongoing evaluation and learning: The experience of continuous iteration and adaptation, together with engagement with researchers in residence, perhaps increased the appetite of project staff for embedding ongoing evaluation into the project. Practically, this led to the decision for the Ecotherapy Lead and assistant to debrief each other after each session and then write up their reflections and observations about each attendee and the group dynamic—with the aim to produce journey maps for individuals and groups, informed by existing observation frameworks [36]. These observations were then discussed with the wider team in weekly one-hour sessions, as part of both a collective sense-making process and a continuous cycle of learning about what was effective about the CYP and staff experiences and how it might be improved. These were captured in exploring ‘golden nuggets’; i.e., anecdotes/quotes from instances of authentically observed connections of everyone involved as learning points to deductively unpack the various dimensions specific (connecting with self, others, animals, nature) and non-specific (therapeutic relationship with practitioners/carers, nature as therapist, and group dynamics) to sessions. This approach constantly surprises with additional, unexpected impacts like hunger for educational content (e.g., encounters with trees/wildlife or setting up video traps for otters); creativity/connectivity of unstructured play and group dynamics (e.g., any self-directed activity in the stream; or modelling clay); or the nuance around engagement with animals (e.g., fascination with eggs; non-judgmental nature of encounters).

Emerging learning was shared throughout the year by inviting visitors to the farm and/or holding weekly meetings, including with other practitioners, to increase referrals/visibility and raise awareness of the farm, as well as with politicians and potential funders. One Young People Peer Researcher from Changing Futures Plymouth (https://shows.acast.com/plymouth-changing-futures-podcast; accessed on 20 April 2025) also collected feedback from a Young Adult Carers Group and staff after completion of the course, which was used for a handout poster for dissemination (available on request from the corresponding author).

More formally, observational and MyCAW data were triangulated to map individual journeys and then collectively ‘make sense’ of the service: what worked for whom and under which circumstances. We felt it was important to begin this process sooner rather than later, before memories of young people and the sessions faded and became confused with more recent events. To augment this real-time data collection and analysis and bring further depth to the work, a Doctorate in Clinical Psychology student was recruited to work on making this process more rigorous in 2025 (to be reported), subject to university ethical review.

## 4. Discussion

The aim of this paper was three-fold in reporting (a) on a pragmatic, opportunistic internal evaluation using a simple, qualitative, appreciative enquiry; (b) sharing methodological reflections on further implementation through co-production to improve the service offer; and (c) discussing the wider implications below.

This is an innovative account showcasing how a public health team has used a Human Learning Systems (https://www.humanlearning.systems/; accessed on 20 April 2025) approach and qualitative/open appreciative enquiry for an internal pilot and service evaluation to demonstrate the potential of ecotherapy services. This resulted in additional funding for a larger two-year evaluation (full findings to be reported in 2025/26) that was co-designed and supported by an innovative research capacity-building approach using embedded researchers and matching them with practitioner researchers and peer researchers.

This methodological reflection of findings from early pilot work and the further development of group-based ecotherapy and its evaluation (going forward) showcases how operational, experiential, and clinical learning can be meaningfully triangulated through a mix of professional, academic, and disciplinary perspectives using a mix of qualitative and quantitative data and using a mix of collective sense-making approaches, with the mutual benefit of helping with the constant and iterative improvement of the service and wider activities in the city and allowing ongoing peer supervision and continuous development to tailor group dynamics safely and in a bespoke fashion while creating a shared, co-produced and emerging understanding of the key ingredients of ecotherapeutic benefit. It combines relevance with rigour in a unique, thick, and contextually rich way that contributes to and reshapes the emerging evidence base of how we relate to nature, self, and each other and how we define what works.

### 4.1. What to Measure?

Due to the small and varied sample size, our initial pilot findings show little shift in self-reported RCADS scores, and looking at emerging MYCaW data, we are hopeful for more meaningful and attributable impact and pragmatic theory of change models (https://aifs.gov.au/resources/practice-guides/what-theory-change, accessed on 20 April 2025). This does raise the question of which data CAMHS is commissioned to collect and if they aremeaningful measures from the perspective of CYP and their carers. Furthermore and more specifically, there is both increasing consent and dissent for common biophilia (innate human affinity) and Stress Reduction Theory findings [38]. In the UK, this is not without controversy, as an exploration of RCADS data within CAMHS services highlights considerable methodological and reporting shortfalls [39]. A recent international review comparing conventional cognitive behavioural therapy with ecotherapy on any-age populations shows similar improvements and remains inconclusive in terms of symptom reduction of anxiety and depression [40].

While evidence for nature-based interventions for adults is emerging on quantitative indicators, including wider conceptualisation of returns beyond cost-effectiveness [41,42], further research is required that is specific to young people [43]. Confirming our findings and reflections, recent qualitative explorations of ecotherapy experiences from both CYP and parents in the UK are more nuanced in reporting reduced negative emotions and anxiety; greater self-esteem, confidence, and social skills; and wider “benefits of engaging in support services that varied from the norm” [44].

From a Human Learning Systems perspective [45], the policy implications would be that if a quantitative outcome measure becomes a target for performance management in the marketisation of public services, it will be gamed (see Campbell’s Law/Goodhart’s Law) [46]. The research implications are similar in that single primary, quantitative outcome measures within the gold standard of Randomised Controlled Trials continue to control for ‘noise’ at the expense of contextual variation and nuance in human behaviour change interventions [47].

This also raises deeper questions about nature vs. nurture and whether the scientific paradigm holds validity and is applicable to the likely complex mix of specific and non-specific treatment effects at play. Existentially, this opens up the argument to link the biophilia hypothesis to the Gaia hypothesis [48] of rediscovering and nourishing the outside within us, with implications for our collective survival as one self-organising, complex system. Some of this is captured in a recent integrative review from Canada, which elaborates on underlying mechanisms and impacts around the concept of ‘social health’, which is defined to include both relationship-enhancing prosocial behaviours and judgments of social connection [49].

Maybe we should stop expecting recovery from symptoms as a main yardstick by which to assess effectiveness and consider talking about outputs rather than outcomes towards a life’s journey (https://www.sopact.com/guides/output-vs-outcome, accessed on 20 April 2025). Complexity-informed approaches challenge us to accept that there might likely be wider, non-linear benefits with regard to human flourishing [50]; that they might be non-measurable by current human standards; and that there might not be an average dose response of exposure to self, others, and nature. Such a comprehensive integrated whole-systems outlook would maybe use more narrative storytelling approaches—such as this one—to evidence transformative impacts and go beyond hierarchical, medicalised, deficit-based models like Maslow’s pyramid of needs (e.g., Max Neef’s Human Scale Development [51]). This would comprise the wider social, environmental, and commercial determinants of health and advances in neuroscience to elucidate what instincts around multisensory integration [52] tell us is a good thing to do (e.g., Sukhvinder Obi’s neuroscience of power, illustrating the ‘we’ over ‘I’ [53]; Lisa Feldman Barrett on Body Budgeting and balancing our energy [54]).

### 4.2. Co-Production and Embedded Research

An international benchmark paper summarises the “know-how” and “art” of conducting knowledge co-production and implementation work in climate science and other environmental sciences [55]. The paper emphasises several contextual factors and principles that we also found to be of relevance, like the benefit of pooling expertise through different services, disciplines and working cultures; and the importance of trust and inclusivity in all processes, with beneficial impact on the salience of the co-production, as well as improved practice. These findings are corroborated by very recent implementation research of nature-based approaches in a CAMHS organisational environment in the Southwest UK, emphasising the culture shift required and the importance of experiential learning [56].

Nationally, co-production through collaboration with service providers, people who use services and carers, commissioners of services, policymakers, and researchers has been promoted by national organisations such as the Think Local Act Personal partnership [57] and the British Psychological Society (https://www.bps.org.uk/news/new-reference-guide-promotes-co-production-young-people-clinical-psychology, accessed on 20 April 2025). Current evidence synthesis more widely suggests there is a scarcity of reporting on meaningful involvement of CYP in designing or evaluating mental health services [58], validating the importance of methodological accounts like this. Similar to our approach, co-design and practice-based reports are, however, emerging, as this recent methodological account from the UK shows, describing an academia–practice–VCFSE partnership working together with girls and young women from disadvantaged backgrounds in Oxford, called ‘Greenspace & Us’ [30].

A knowledge-to-practice gap is acknowledged in literature reviews of knowledge utilisation in local government, with a range of barriers explored and a preference stated by practitioners for interactional, face-to-face engagement with researchers [59,60]; embedding research and engaging peers [61] and practitioners in co-design and evaluation is a direct way of achieving this.

The recently launched programme of thirty HDRC programmes in the UK (https://www.nihr.ac.uk/about-us/what-we-do/working-with-partners/local-authorities/health-determinants-research-collaborations, accessed on 20 April 2025) must be seen as a large-scale investment to redress the balance. While the existing knowledge base about developing research capacity in local government through any means is scant at present, evidence from public health [62] and health [63] research suggests the potential value of embedded research. The Plymouth HDRC programme addresses the transferability of this approach from health to local government more widely and will support the much-needed understanding of how knowledge production and transfer can best be developed [64].

This project is concerned with developing research capacity through embedding researchers in practice settings, but it also has the significant element of linking them to practitioners who will remain in practice after the project ends to champion and support sustainable organisational change. The research policy implications would be that if they invest in people and place through matched-funded and pooled capacity-building like this, academics are prevented from having to conduct largely extractive, project-based data collection approaches. Instead, feedback loops of learning are more continuous, and agendas and processes are more jointly owned. This might more likely lead to mutual and sustainable pathways to impact. The potential of ‘practitioner research’ has been promoted by the NIHR School of Social Care Research [65], and this is gaining traction internationally by public health colleagues [66]. Others have cautioned that these claims should be linked to more rigorous research approaches [67], something we attempt to bridge by matching them up with senior academics and the joint supervision of Doctorate in Clinical Psychology students, who will jointly report in due course. This project will provide a carefully evaluated test bed for claims about practitioner and embedded research, and about the feasibility of making it robust enough to be the basis for wider knowledge transfers.

## 5. Conclusions

The Southwest of the UK is nationally leading around social prescribing (for adults, which is better developed than young people currently; https://www.plymouth.ac.uk/research/primarycare/social-prescribing, accessed on 20 April 2025), as well as in partnership with others on green/blue prescribing (https://www.ecehh.org/research/nature-prescription-handbook/, accessed on 20 April 2025).

This account gives a rare report of innovating locally with a bespoke, cross-system partnership that is and will be reporting rapidly, and it is owned and co-designed by/with/for people. It is an emerging example of ‘anchor organisations’ working together (i.e., big local employers with a shared civic duty; https://haln.org.uk/case-studies/researchers-in-residence, accessed on 20 April 2025), an important educational contribution to addressing the current unstable and tipping state of ‘ecological overshoot’, where human consumption outpaces the Earth’s ability to regenerate [68].

## Figures and Tables

**Table 1 ijerph-22-00720-t001:** RCADS before and after scores.

Sample	T Score Before	T Score After	Percentage Change
Male/Age 12	68	50	26%
Male/Age 11	71	54	24%
Male/Age 11	61	51	16%
Male/Age 10	37	31	16%
Male/Age 13	64	57	11%
Male/Age 11	77	70	9%
Male/Age 11	64	64	0%
Male/Age 13	54	>80	−48%

## Data Availability

More detailed draft reports on the service evaluations can be requested from the corresponding author.

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
