# Peer review of "Pilot Testing an Ecotherapy Program for Adolescence: Initial Findings and Methodological Reflections"

_ijerph, 2025, doi:10.3390/ijerph22050720_

Round 1
Reviewer 1 Report
Comments and Suggestions for Authors
At first glance it seems to seek to contribute to the body of literature on the use of co-design principles, community-engaged programming, and implementation research. This paper covers a lot of ground, some of which is covered in other papers that address participatory-based research, patient-reported outcomes, and implementation research, more of which should be included in the introduction. The following papers need not be included specifically, but are examples of the broader scope of literature that contributes to the arguments made in this paper.
Additional references to work on
Co-production Djenontin INS, Meadow AM. The art of co-production of knowledge in environmental sciences and management: lessons from international practice. Environmental management. 2018;61(6):885-903.
Qualitative Research: Pope C, Mays N. Qualitative Methods in Health Research. Qualitative Research in Health Care2006. p. 1-11.
Implementation Science: Bauer MS, Kirchner J. Implementation science: What is it and why should I care? Psychiatry research. 2020;283:112376.
Over the course of the paper the authors comment on describing the development of a new program through co-design, knowledge gaps in several areas, from nature-based interventions to evaluation methodology. Subsequently, they turn their attention to criticizing current practices in evaluation. These twists in objectives are not clear from the introduction and obscure the message about the purpose, methods, and outcomes of the project conducted by the authors. The manuscript can be improved by greater attention to the clarifying the methodology and emphasizing the outcomes of the project at hand. Of primary concern is for the authors to disentangle the purpose of the paper – is it a report on the feasibility of a pilot? Is it a case study report? Is it a criticism of current methodologies? The paper can do all three but needs more structure so the reader can identify the goals.
1. Introduction:
The introduction emphasizes the purpose of the paper as reporting on two separate activities -a pilot/feasibility project and a co-development planning period- to assess methods to expedite the knowledge to practice timeline for programs to improve the mental health of children. However, the authors do not return to this objective in the Discussion.
The use of terminology in the final paragraph of the Introduction (“…reports on a small-scale pilot and feasibility … providing the case study…” needs to be clarified and made consistent with use in the Materials and Methods section.
It is not clear why it was important that the study be focused on ecotherapy or whether the practices and concerns expressed in the paper could apply to any form of programming intended to improve the mental health of children and youth. Was this simply a convenient program in which to study the process of co-design? The brief paragraph about urban green space (lines 58 through 64) do not explain concerns about a knowledge gap in the effectiveness of nature-based interventions, issues that are raised later in the paper. As written, it seems that the fact that the project focused on a program that used nature-based interventions was simply incidental to the study. If the evaluation of nature-based interventions is central to the current or future stories to be told, that should be clarified in the introduction.
2. Methods:
The paper reports on two separate, but sequential components of program development, the first being a pilot/feasibility project, the second being the co-design initiative. The methodology for assessing evaluation methods is not clearly described.
To help the reader parse out what components belong to which part of the project, labeling them as 2.1 Pilot/Feasibility Study and 2.2 Ecotherapy Co-Design Process would be helpful. Also, the introduction references “pilot and feasibility” whereas Materials and Methods leads with “Feasibility Study…”, please be consistent in the use of terminology.
Line 129: It is unclear why the reference to the one participant is relevant.
Lines 132-133 Comment about mixed gender and smaller groups seems more appropriate for results and should be accompanied with an explanation of why the staff felt this was the case.
Lines 137-139 It is not clear if all cohorts visited both sites or only one. If a cohort visited only one site, then any differences that might be the result of differences in exposure should be discussed in the results; if there were no differences due to location, then that should be reported also.
Line 143. Presumably the Appreciative Enquiry prompts were directed at CYPs, please specify.
Line 155 As a signpost to the reader, the switch to discussing the co-design phase should be a new subsection (e.g. 2.2…).
3. Results
Line 188. Either here or earlier, briefly describe what a Human Learning Systems approach is. Also give a reference describing the approach in this sentence as it is such an important concept.
Line 198 and following. Be specific about to which aspect of the study the results you are describing pertain. Does section 3.1 refer to the pilot/feasibility study or the co-design project?
Line 259 Again, be consistent with terminology. Does “Continued Iteration” refer to the co-design process? If so, refer to this section as part of the co-design process? If this is an iteration of the co-design process where did reporting of the initial results of the co-design process begin?
Lines 269-283 – Does this refer to work with groups in the pilot/feasibility phase or during co-design meetings?
Providing a time-line that illustrates when different types of sessions were held might be helpful.
Line 299 The authors use the term “fox-walking” here and earlier in the paper, please define or describe.
Lines 304-315 Not clear whether this on-boarding was for the pilot/feasibility study, the co-design process, or both.
Lines 323-315 Reference to use of the Hayloft in 2024 makes it unclear whether this is part the pilot/feasibility study (I thought that was all pre-COVID) versus part of the co-design process (2024).
Lines 320-323 Cite websites for the programs/locations referenced.
4. Discussion
Lines 417-429 This extended paragraph identifies the need to use the right “tool” to assess outputs versus outcomes (see https://www.sopact.com/guides/output-vs-outcome). However, in light of the usually limited budgets and time allocated for evaluation, service organizations often do not have the resources needed to do the comprehensive, narrative, assessments suggested by the authors. That isn’t to say they are not needed or that the measures used shouldn’t be changed, but what are program managers to do. I would prefer to see this paragraph de-emphasize the negatives and suggest a more constructive and direct suggestion for change. For example, if one were to apply a Theory of Change model (https://aifs.gov.au/resources/practice-guides/what-theory-change) to developing an evaluation model, would it be possible to identify some acute changes that are more feasible for agencies to monitor?
Author Response
Comments 1 and Suggestions for Authors
At first glance it seems to seek to contribute to the body of literature on the use of co-design principles, community-engaged programming, and implementation research. This paper covers a lot of ground, some of which is covered in other papers that address participatory-based research, patient-reported outcomes, and implementation research, more of which should be included in the introduction. The following papers need not be included specifically, but are examples of the broader scope of literature that contributes to the arguments made in this paper.
Response1: Thank you for this fair, overall assessment, and concrete steer with regards to pitch, aims/objectives, overall flow, and consistency of language and messaging. These perfectly align with reviewer2 comments, and we have taken liberty to merge these as indicated specifically in our responses to both reviewers respectively and highlighted through tracked changes in the updated manuscript. Thank you also for concrete, relevant additional references below, one of which we have added to the introduction – noting that none of us are academic experts in the topic area of ecotherapy, yet.
Comments2: Additional references to work on
Co-production Djenontin INS, Meadow AM. The art of co-production of knowledge in environmental sciences and management: lessons from international practice. Environmental management. 2018;61(6):885-903.
Response2: Thank you for pointing out this excellent paper, which we have highlighted at the top of the 4.2. co-production section of the discussion (lines 483-489) as an international benchmark paper on the approach we seek to emulate.
Comments3: Over the course of the paper the authors comment on describing the development of a new program through co-design, knowledge gaps in several areas, from nature-based interventions to evaluation methodology. Subsequently, they turn their attention to criticizing current practices in evaluation. These twists in objectives are not clear from the introduction and obscure the message about the purpose, methods, and outcomes of the project conducted by the authors. The manuscript can be improved by greater attention to the clarifying the methodology and emphasizing the outcomes of the project at hand. Of primary concern is for the authors to disentangle the purpose of the paper – is it a report on the feasibility of a pilot? Is it a case study report? Is it a criticism of current methodologies? The paper can do all three but needs more structure so the reader can identify the goals.
Response3: Thank you very much for this encouragement and following these, and the concrete suggestions from reviewer 2 to list the aims/purposes more clearly and upfront for better structure and flow, we have added these suggested three-fold aims at the bottom of the introduction (lines 97-101), and revisit these at the beginning of the discussion (lines 399-402); see also comments/response4 below.
- Introduction:
Comments4: The introduction emphasizes the purpose of the paper as reporting on two separate activities -a pilot/feasibility project and a co-development planning period- to assess methods to expedite the knowledge to practice timeline for programs to improve the mental health of children. However, the authors do not return to this objective in the Discussion.
Response4: Thank you very much for this steer to top and tail messaging to improve the clarity of the overall manuscript. We have added clearer aims and objectives in the introduction to then revisit in the discussion.
Comments5: The use of terminology in the final paragraph of the Introduction (“…reports on a small-scale pilot and feasibility … providing the case study…” needs to be clarified and made consistent with use in the Materials and Methods section.
Response5: Thank you for this helpful suggestion to use consistent language throughout in line with the clarified aims – as suggested by both reviewers. We have used more service-based language (‘pilot’, ‘evaluation’, ‘co-design’, ‘co-production’, ‘learning’) rather than academic language (‘feasibility’, ‘intervention’, ‘case study’, ‘research’) to make clearer that this was always more of a practice-led, rather than academia-led programme (i.e. academics coming to this only recently to assist with training, co-design, reporting, and helping with further formal research in future).
Comments6: It is not clear why it was important that the study be focused on ecotherapy or whether the practices and concerns expressed in the paper could apply to any form of programming intended to improve the mental health of children and youth. Was this simply a convenient program in which to study the process of co-design?
Response6: Yes, as described in the setting section at the beginning of the materials/method section (line 130ff) this project sits within several currently emerging projects across a range of topics related to the social/commercial/environmental determinants of health, within the wider 5-year programme of embedded research seeking to build capacity for actionable learning across our City and region. As per your helpful steer, we have clarified the triple aim of the paper to include discussion of specific (to ecotherapy, and mental health services) and wider reflections (cultures of public service delivery, related academic paradigms) through participatory, embedded approaches. We hope this makes our ambitions for transferable learning and messaging clearer.
Comments7: The brief paragraph about urban green space (lines 58 through 64) do not explain concerns about a knowledge gap in the effectiveness of nature-based interventions, issues that are raised later in the paper. As written, it seems that the fact that the project focused on a program that used nature-based interventions was simply incidental to the study. If the evaluation of nature-based interventions is central to the current or future stories to be told, that should be clarified in the introduction.
Response7: Thank you for this concrete suggestion to clarify here, and better link through to the discussion, which we now seek to address by clarifying the aims (see above and reviewer2 comments). As you helpfully suggest, we have added a sentence to introduce a potential knowledge gap in terms of qualitative process indicators, theory around mechanisms of change, and consideration of wider outcome measures (now lines 67-69).
- Methods:
Comments8: The paper reports on two separate, but sequential components of program development, the first being a pilot/feasibility project, the second being the co-design initiative. The methodology for assessing evaluation methods is not clearly described.
Response8: Thank you for this feedback, which aligns with reviewer1 comments. We have clarified the aims throughout, seeking to top and tail this framing better. We have added a paragraph on the analysis methods (lines 146-150)
Comments9: To help the reader parse out what components belong to which part of the project, labeling them as 2.1 Pilot/Feasibility Study and 2.2 Ecotherapy Co-Design Process would be helpful. Also, the introduction references “pilot and feasibility” whereas Materials and Methods leads with “Feasibility Study…”, please be consistent in the use of terminology.
Response9: Thank you for this suggestion to improve the structure and make it easier for readers to navigate. We have changed the headings as suggested. We have also distinguished these two parts/aims of the report more clearly by using consistent terms throughout the manuscript
Comments 10: Line 129: It is unclear why the reference to the one participant is relevant.
Response 10: Thank you and we have deleted this sentence, which tried to make the distinction between face-to-face and online meetings, due to the logistics/uncertainties during the emerging pandemic (lines 138-9).
Comments 11: Lines 132-133 Comment about mixed gender and smaller groups seems more appropriate for results and should be accompanied with an explanation of why the staff felt this was the case.
Response11: Thank you and we have deleted this incidental finding here (lines 142-3) partly to save space, and as gendered group dynamics are to be elaborated more robustly in future reporting. We have learned so much more about this since.
Comments 12: Lines 137-139 It is not clear if all cohorts visited both sites or only one. If a cohort visited only one site, then any differences that might be the result of differences in exposure should be discussed in the results; if there were no differences due to location, then that should be reported also.
Response12: Thank you for spotting this, and apologies for this confusion. All the early internal evaluation work was conducted on one single site (green prescribing, at Poole Farm). We have clarified this accordingly (now line 131), and will report in future about emerging, and relevant comparison between green and blue sites.
Comments13: Line 143. Presumably the Appreciative Enquiry prompts were directed at CYPs, please specify.
Response13: Thank you for this query for important clarification. We have added that these were directed at both CYPs and their carers/parents (depending on preference and availability) (line151, 155, 159).
Comments14: Line 155 As a signpost to the reader, the switch to discussing the co-design phase should be a new subsection (e.g. 2.2…).
Response 14: Thank you, and as per above response, we have used your suggestion to add numbering to the headings, hoping to make this easier for the readers to navigate.
- Results
Comments 15: Line 188. Either here or earlier, briefly describe what a Human Learning Systems approach is. Also give a reference describing the approach in this sentence as it is such an important concept.
Response15: Thank you for this encouragement to elaborate further and we have added a weblink (https://www.humanlearning.systems/), and a recent academic reference here and elsewhere in the discussion (https://doi.org/10.1080/09540962.2025.2456120). Following reviewer2 suggestions we have shortened and added this to the summary paragraph at the beginning of the discussion.
Comments 16: Line 198 and following. Be specific about to which aspect of the study the results you are describing pertain. Does section 3.1 refer to the pilot/feasibility study or the co-design project?
Response 16: Thank you for this helpful suggestion for better signposting, which we have added here (line 215).
Comments 17: Line 259 Again, be consistent with terminology. Does “Continued Iteration” refer to the co-design process? If so, refer to this section as part of the co-design process? If this is an iteration of the co-design process where did reporting of the initial results of the co-design process begin?
Response17: Thank you for this helpful pointer to better signpost throughout the manuscript. We have deleted the word ‘continued iteration’ (line 281) and added a sentence to clearer mark the distinction between early evaluation and later co-design.
Comments 18: Lines 269-283 – Does this refer to work with groups in the pilot/feasibility phase or during co-design meetings? Providing a time-line that illustrates when different types of sessions were held might be helpful.
Response 18: Thank you for suggesting clarification here, we have added a time stamp here (line 291) to make clearer that this refers to the most recent co-design events in 2024.
Comments19: Line 299 The authors use the term “fox-walking” here and earlier in the paper, please define or describe.
Response19: Thanks for asking for more detail and we have added at first mention (lines 197-200) a brief description of the activity and purported core mechanisms/intentions (mindfulness, sensory attunement, and trusting self/others).
Comments20: Lines 304-315 Not clear whether this on-boarding was for the pilot/feasibility study, the co-design process, or both.
Response20: Thank you for this query, and on-boarding was only developed as a referral process in 2024 so this relates to developing the wider service offer after the co-design events. We have added content (line 327) to clarify, hoping to make this clear.
Comments21: Lines 323-315 Reference to use of the Hayloft in 2024 makes it unclear whether this is part the pilot/feasibility study (I thought that was all pre-COVID) versus part of the co-design process (2024).
Response22: Thank you for raising this in terms better clarifying timelines. We have added a reference to timelines in these sections.
Comments23: Lines 320-323 Cite websites for the programs/locations referenced.
Response 24: Thank you for encouraging us to showcase wider nature-based activities in the City; we have added relevant websites to the National Trust, National Marine Park, and Green Minds Plymouth.
- Discussion
Comments24: Lines 417-429 This extended paragraph identifies the need to use the right “tool” to assess outputs versus outcomes (see https://www.sopact.com/guides/output-vs-outcome). However, in light of the usually limited budgets and time allocated for evaluation, service organizations often do not have the resources needed to do the comprehensive, narrative, assessments suggested by the authors. That isn’t to say they are not needed or that the measures used shouldn’t be changed, but what are program managers to do. I would prefer to see this paragraph de-emphasize the negatives and suggest a more constructive and direct suggestion for change. For example, if one were to apply a Theory of Change model (https://aifs.gov.au/resources/practice-guides/what-theory-change) to developing an evaluation model, would it be possible to identify some acute changes that are more feasible for agencies to monitor?
Response24: Thank you for these helpful and well-considered observations. We agree that there is a balance to be struck with pragmatic, good-enough accounts and deeper explorations of theory of change models. We have included your helpful references and have softened the messaging to be more constructive in respective sections. We have also considered this together with similar observations from reviewer2, and added a recent, relevant references.
Reviewer 2 Report
Comments and Suggestions for Authors
This is an interesting manuscript on an important topic. That stated, there are some issues that I suggest the authors address to improve the overall readability and quality of the manuscript.
Title
- I suggest you shorten the title so that it more accurately reflects the topic. The title seems like it is trying to communicate too much (i.e., the part: 'Outside-Within' does not make sense and will mean little to the reader), and this risks losing reader interest. I suggest you refocus your title to something like: Pilot testing an ecotherapy program for adolescence: Initial findings and methodological reflections.
Introduction
- line 78: define 'third setor services' as this is an unfamilar term in North America
- line 119: should this read as "reported" instead of "reporting..."?
- line 95-100: The purpose statement is unclear. There is clearly a desire to present some preliminary findings from the children participants as well as methodological reflections. I suggest you revise this section to clearly state the goals of the paper using (a) and (b) format. For example: The purpose of the paper was to (a) provide preliminary data for the children who participated in the intervention, and (b) report the methodological reflections made by the team that informed the design and implementation of the intervention.
Method
- line 155: the sentence including "...fairly successully..." was awkward and needs revision.
- how were the qualitative results analyzed? By whom?
- was the study reviewed by an ethics board? How was consent to participate achieved?
Results
- The first paragraph of the Results section seems misplaced, and may fit better in the Discussion.
- If the purpose statement is revised, the presentation of the findings will flow better.
- line 335: the sentence "... the extent to which they are bothering them." needs to be revised. It is unclear what 'they' and 'them' is referring to.
Discussion
- Provide more references to existing research that supports ecotherapy for children and the benefits (or challenges) of working in multidisciplinary teams.
Author Response
Comments and Suggestions for Authors
This is an interesting manuscript on an important topic. That stated, there are some issues that I suggest the authors address to improve the overall readability and quality of the manuscript.
Title
- Comments1: I suggest you shorten the title so that it more accurately reflects the topic. The title seems like it is trying to communicate too much (i.e., the part: 'Outside-Within' does not make sense and will mean little to the reader), and this risks losing reader interest. I suggest you refocus your title to something like: Pilot testing an ecotherapy program for adolescence: Initial findings and methodological reflections.
Response1: Thank you for this overall assessment on the relevance of the manuscript and topic. Thank you for helping us clarify the pitch, aims/objectives and overall flow which perfectly align with reviewer1 comments, including the suggestion to shorten the title, which we have changed accordingly.
Introduction
- Comments 2: line 78: define 'third setor services' as this is an unfamilar term in North America
Response2: Thank you for helping us make this more portable to an international audience, we have changed this expression, to hopefully conceptualise this more clearly/inclusively by introducing the acronym of VCFSE (Voluntary and Community, Faith and Social Enterprise Sector), and are using this more consistently throughout the manuscript.
- Comments3: line 119: should this read as "reported" instead of "reporting..."?
Response3: Yes thank you for spotting this grammatical error, we have changed it accordingly (now line 129, in updated manuscript with track changes).
- Comments4: line 95-100: The purpose statement is unclear. There is clearly a desire to present some preliminary findings from the children participants as well as methodological reflections. I suggest you revise this section to clearly state the goals of the paper using (a) and (b) format. For example: The purpose of the paper was to (a) provide preliminary data for the children who participated in the intervention, and (b) report the methodological reflections made by the team that informed the design and implementation of the intervention.
Response4: Thank you for this concrete suggestion which we have used to clarify the aims of the paper at the bottom of the introduction (lines 97-101). Following reviewer1 comments to introduce a three-fold framing, we also re-visit those at the beginning of the discussion section, for better topping and tailing, structure of messaging, and overall flow (lines 399-402).
Method
- Comments5: line 155: the sentence including "...fairly successully..." was awkward and needs revision.
Response5: Thank you for sense-checking and spotting this typo. This is now simplified and divided into several sentences to hopefully make this point clearer (lines 171-173).
- Comments6: how were the qualitative results analyzed? By whom?
Response6: Thank you for pointing out this omission of a more detailed description regarding analysing and reporting. We have added a paragraph in the methods section (lines 146-150), outlining the process and roles of co-authors (which are also described in the author contributions at the bottom of the paper).
- Comments7: was the study reviewed by an ethics board? How was consent to participate achieved?
Response7: Thank you for this, and see institutional review board statement at the bottom of the paper (lines 570-577), that hopefully gives reassurance that ethical principles were applied. We also want to clarify that this manuscript reports on internal service evaluation, clinical audit, co-design, and quality improvement (with a plan for formal research in future: see UK guidance here: https://www.hra-decisiontools.org.uk/research/docs/DefiningResearchTable_Oct2022.pdf).
We have changed the language, and terms used throughout the manuscript, hoping to make this distinction clearer.
Results
- Comments8: The first paragraph of the Results section seems misplaced, and may fit better in the Discussion.
Response8: Thank you for this helpful suggestion, which aligns with reviewer1 comments – asking for more elaboration here. We have partly deleted, moved content back to the discussion, and revised this section (lines 403-409).
- Comments9: If the purpose statement is revised, the presentation of the findings will flow better.
Response9: See above response4, we hope this reads better overall, also aligning with the adapted framing and flow, hopefully helped through clarified aims and tighter messaging.
- Comments10: line 335: the sentence "... the extent to which they are bothering them." needs to be revised. It is unclear what 'they' and 'them' is referring to.
Response10: Thank you for spotting this, and we have changed the wording to make this clearer in terms of being goal-oriented and person-centered (lines 357-359).
Discussion
- Comments11: Provide more references to existing research that supports ecotherapy for children and the benefits (or challenges) of working in multidisciplinary teams.
Response11: Thank you for your suggestion to reference more specific literature in the discussion, which we have done by adding 6 additional references. In line with suggestions from reviewer1, we also sought to soften and tighten the messaging overall (deleting various content).
Round 2
Reviewer 2 Report
Comments and Suggestions for Authors
I think that the authors addressed the issues I raised in their revision. There are a few typos (e.g., "...no-systematically analyzed...") which I assume will be corrected during the production process.
Author Response
Comment 1: I think that the authors addressed the issues I raised in their revision.
Response1: Thank you very much for accepting the major revisions submitted, and for the close-reading and time given so generously. We are very grateful and look forward to publishing this.
Comment 2: There are a few typos (e.g., "...no-systematically analyzed...") which I assume will be corrected during the production process.
Response 2: Thank you for spotting this typo which we have corrected, we also did another proof-read and found other minor typos, see track changes in revised manuscript. We have also tidied up the additional (now 7) literature references, and updated those in-text as well as the full reference list at the bottom of the paper.